

# SERGHEI v2.0: introducing a performance-portable, high-performance three-dimensional variably-saturated subsurface flow solver (SERGHEI-RE)

Zhi Li[1], Gregor Rickert[2], Na Zheng[1], Zhibo Zhang[1], Ilhan Özgen-Xian[2], and Daniel Caviedes-Voullième[3,4]

[1]College of Civil Engineering, Tongji University, Shanghai, China
[2]Institute of Geoecology, Technische Universität Braunschweig, Brunswick, Germany
[3]Simulation and Data Lab Terrestrial Systems, Jülich Supercomputing Centre, Forschungszentrum Jülich, Jülich, Germany
[4]Institute of Bio- and Geosciences Agrosphere (IBG-3), Forschungszentrum Jülich, Jülich, Germany

**Correspondence:** Zhi Li (zli90@tongji.edu.cn)

**Abstract.** Advances in high-performance parallel computing have significantly enhanced the speed of large-scale hydrological simulations. However, the diversity and rapid evolution of available computational systems and hardware devices limit model flexibility and increase code maintenance efforts. This paper introduces SERGHEI-RE, a three-dimensional, variably saturated subsurface flow simulator within the SERGHEI model framework. SERGHEI-RE adopts the Kokkos-based portable paral-

lelization framework of SERGHEI, which facilitates scalability and ensures performance portability on various computational devices. Moreover, SERGHEI-RE provides options to solve the Richards Equation with iterative or non-iterative numerical schemes, enhancing model flexibility under complex hydrogeological conditions. The solution accuracy of SERGHEI-RE is validated using a series of benchmark tests, ranging from simple infiltration problems to practical hydrological, geotechnical and agricultural applications. The scalability and performance portability of SERGHEI-RE are demonstrated on a desktop

workstation, as well as on multi-CPU and multi-GPU clusters, indicating that SERGHEI-RE is an efficient, scalable and performance portable tool for large-scale subsurface flow simulations.

## 1 Introduction

In 2022, the Frontier supercomputer with 1.206 exaFLOPS computing power went online, marking the beginning of the exascale computing era, which will promote previously unfeasible advanced modeling studies in various fields (Chang et al.,

2023). Exascale computing will also benefit computational hydrologists who study complex multiscale hydrological processes at catchment to regional scales. Such studies often require the numerical simulation of variably-saturated subsurface flow, which can be computationally intensive due to the nonlinear governing equations and the wide ranges of spatial and temporal scales involved (Farthing and Ogden, 2017; Mao et al., 2021; Paniconi and Putti, 2015; Zha et al., 2019).

Most modern physically-based integrated hydrological models use the Richardson–Richards equation (Richardson, 1922;

Richards, 1931) (hereinafter referred to as *Richards equation* for historical reasons) to describe variably-saturated subsurface flow, see, for example, Paniconi and Putti (2015). Here, a critical challenge is that due to the non-linearity of the equation itself





and the required closure relationships for soil-water retention and soil hydraulic conductivity, no numerical scheme achieves accuracy, efficiency and robustness simultaneously (Farthing and Ogden, 2017; Li et al., 2024). Additional challenges regarding efficient time stepping, treatment of boundary conditions, and estimation of the interfacial conductivity further complicate the

design or selection of numerical Richards solvers (D'Haese et al., 2007; El-Kadi and Ling, 1993; Lai and Ogden, 2015; Paniconi and Putti, 1994; Zha et al., 2016).

Subsurface flow simulations in the hydrological context are often performed over spatial scales that horizontally go from hillslope to the catchment scale and vertically up to hundreds of metres into the subsurface (Condon et al., 2020; Özgen-Xian et al., 2023). The temporal scales of such simulations are often a couple of years to decades. These spatio-temporal scales are

usually magnitudes larger than the required resolutions required to accurately represent the involved hydrological processes. Although relatively coarse grid resolutions are generally accepted for modeling fully saturated groundwater in the deeper soil layers, in the unsaturated zone, grid resolution is often significantly refined below a metre to capture the local flow field (e.g., sharp infiltration fronts). The time step size is reduced accordingly to maintain convergence and stability, see Caviedes-Voullième et al. (2013); Hou et al. (2022). This results in a heavy computational burden. Some existing models address this

issue by assuming one-dimensional (1D) flow in the unsaturated zone, but this assumption leads to biased model predictions when lateral flow is significant (Mao et al., 2021; Shen and Phanikumar, 2010).

Instead of model simplification, high-performance computing (HPC) through massive parallelization provides the computational capabilities to simulate high-resolution, full-dimensional subsurface flow. Richards solvers in most modern hydrological models have parallel computing capabilities, for example, Amanzi-ATS (Coon et al., 2019), PFLOTRAN (Hammond et al.,

2014), ParFlow (Kollet and Maxwell, 2006), Hydrus2D/3D (Šimůnek et al., 2016), and RichardsFOAM (Orgogozo et al., 2014). A foreseeable challenge for many of these models—especially in the exascale era—is the plethora of parallel programming models and their rapid evolution, especially of GPU solutions. Well established standards, such as OpenMP and MPI, are highly portable across HPC systems. However, GPU programming models (CUDA, HIP) are hardware dependent. In order to allow for models to be deployed on as many HPC systems as possible, maintaining multiple programming models in the

source code that target different HPC architectures becomes necessary. While this results in complex code and increased software maintenance efforts (technical debt), not providing these capabilities will significantly limit the model's applicability in "real world" use cases (Hokkanen et al., 2021).

A key challenge that arises when maintaining multiple programming models in the source code is *performance-portability*, that is to say the ensurance that the parallel performance and scalability of the code is preserved across different HPC hardware.

To the best of our knowledge, despite its importance, performance-portability has not yet been the focus of the computational hydrology community. An exception is the ParFlow model that proposed to use an embedded Domain Specific Language (eDSL) to achieve performance-portability, but its performance has not been demonstrated on a wide range of hydrological applications and computational systems (Hokkanen et al., 2021).

The Simulation EnviRonment for Geomorphology, Hydrodynamics, and Ecohydrology in Integrated form (SERGHEI) is an

open-source multi-physics modeling framework for environmental, hydrological and earth system simulations that aims to address the issue of performance-portability in computational hydrology. Performance-portability is achieved through the Kokkos





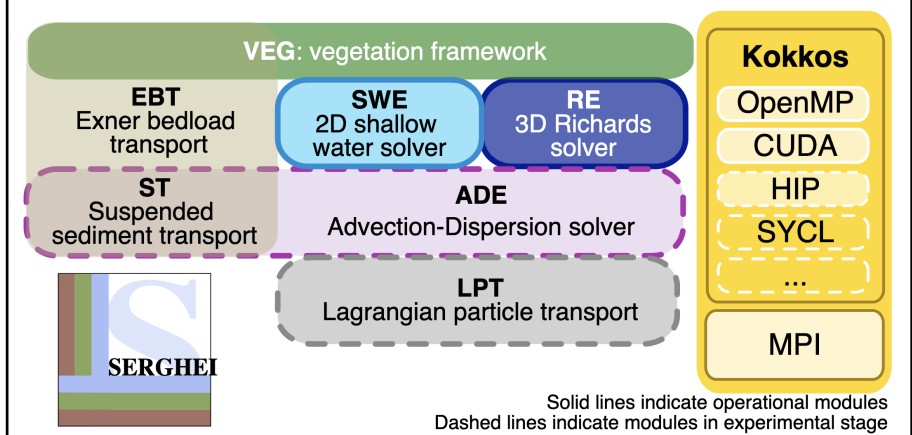

**Figure 1.** An overview of the SERGHEI model components modified from Caviedes-Voullième et al. (2023). The shallow water (SWE) solver has been released. The Richards solver is described in this manuscript. Other components are under development.

programming model, which provides flexibility when switching between different computational platforms, by abstracting hardware-dependent code, and thus, enhancing development productivity and facilitating maintenance (Trott et al., 2022). This means that the same code can be compiled to allow for CPU-parallelization (e.g., OpenMP) and GPU-parallelization (e.g., CUDA), and thus, there is no need to maintain different programming models in the SERGHEI code base. In this manuscript, we introduce SERGHEI-RE, the Richards Equation-based variably-saturated subsurface flow module of SERGHEI.

In the following, the accuracy, robustness, scalability, and portability of the SERGHEI-RE module is demonstrated through a series of numerical tests ranging from simple, idealized problems to large-scale, realistic problems (Section 4 and 5). SERGHEI-RE is designed both as a stand-alone variably-saturated subsurface flow model and as part of the integrated surface-subsurface flow simulator under the SERGHEI framework. In this manuscript, we focus on demonstrating its performance as a Richards solver only.

## 2    Model Description

### 2.1    An Overview of SERGHEI

SERGHEI features a modular design that allows different modules that represent different ecohydrological/hydraulic processes to be dynamically coupled at compile time. The modular architecture of SERGHEI and its existing modules are sketched in Fig. 1. At the time of writing, different modules are at different stages of development. The most mature module of SERGHEI is its fully-dynamic shallow water equations-based surface hydrology solver, released as SERGHEI-SWE in Caviedes-Voullième et al. (2023). The subsurface flow solver SERGHEI-RE is introduced in the following. The coupling of SERGHEI-SWE and SERGHEI-RE to obtain an integrated hydrological solver is currently functional but requires further verification of correctness, numerical accuracy, and performance-portability. A general advection–dispersion equation solver (SERGHEI-ADE) is



functional and serves as a base for the suspended sediment transport module (SERGHEI-ST). Lagrangian particle transport (SERGHEI-LPT) and an Exner equation-based bedload transport module (SERGHEI-EBT) are under development. Finally, the infrastructure to couple SERGHEI with ecosystem models (SERGHEI-ECO) is in early stages of development.

Within this framework, SERGHEI-RE plays a key role in the redistribution of water that drives the other processes discussed above. The performance of this module is thus critical for the entire modelling framework. SERGHEI-RE is equipped with two key features that address these performance expectations: (i) SERGHEI-RE allows the user to choose between an iterative and a non-iterative numerical scheme for solving the Richards equation, which enables efficient time integration on HPC systems under various hydrogeological conditions (Section 2) and (ii) SERGHEI-RE inherits the Kokkos-based code structure of SERGHEI-SWE, which helps to achieve performance-portability on a variety of computational backends (Section 3). Both

features aim at enhancing the adaptivity and flexibility of SERGHEI-RE in handling various simulation scenarios and HPC systems.

## 2.2 Numerical methods

In this section, we describe the numerical schemes applied to solve the Richards equation. The generalized (i.e., considering the compressibility effect when the soil is fully saturated) three-dimensional (3D) Richards equation is presented in Eq. (1)

in its mixed form, where $S_s$ $[L^{-1}]$ is the specific storage, $h$ $[L]$ is pressure head, $\theta$ $[-]$ is water content, $\phi$ $[-]$ is porosity, $\boldsymbol{q}$ $[LT^{-1}]$ is groundwater flux, which is calculated through a generalized Darcy's Law (Eq. 2). $\boldsymbol{K}$ $[LT^{-1}]$ is the hydraulic conductivity, $z$ $[L]$ is elevation, $S(x,y,z,t)$ $[T^{-1}]$ represents the source/sink terms.

$$\frac{S_s\theta}{\phi}\frac{\partial h}{\partial t} + \frac{\partial\theta}{\partial t} - \nabla\cdot\boldsymbol{q} = S(x,y,z,t) \tag{1}$$

$$\boldsymbol{q} = \boldsymbol{K}\nabla(h-z) \tag{2}$$

The popular Mualem-van Genutchen (VG) model (Eq. 3-5) is used for describing the soil-water retention characteristics (Mualem, 1976; van Genuchten, 1980):

$$s(h) = \left(1+|\alpha h|^{\tilde{n}}\right)^{-m} \tag{3}$$

$$\theta(h) = \theta_r + (\phi-\theta_r)s(h) \tag{4}$$

$$K(h) = K_s s(h)^{\frac{1}{2}}\left[1-\left(1-s(h)^{\frac{1}{m}}\right)^m\right]^2 \tag{5}$$

where, $s$ $[-]$ is water saturation, $K_s$ $[LT^{-1}]$ is saturated hydraulic conductivity, $\theta_r$ $[-]$ is residual water content, $\alpha$ $[L^{-1}]$ and $\tilde{n}$ $[-]$ are the soil parameters and $m = 1 - 1/\tilde{n}$.

SERGHEI-RE provides two numerical schemes to solve Eqs. (1-5). The default option is the predictor-corrector (PC) scheme proposed by Kirkland et al. (1992). An alternative option is the modified Picard (MP) scheme proposed by Celia et al. (1990). The MP scheme has been widely adopted in popular Richards solvers (Šimůnek et al., 2013), which falls, more generally, under



the umbrella of fully implicit solvers which may also use other linearization strategies such as Newton schemes (Kuffour et al., 2020). The MP is an iterative scheme that first solves the linearized Richards equation in its head form as:

$$C_h \frac{\partial h}{\partial t} - \nabla \cdot \boldsymbol{q} = S(x, y, z, t), \tag{6}$$

where, $C(h) = \partial\theta/\partial h$. Then, the linearization error is corrected iteratively with the correction derived from truncated Taylor series. The main drawback of the MP scheme is that convergence of the linearization iterations is not guaranteed. Specifically,

cases using the modified Picard iteration struggle to converge if abrupt changes in soil moisture occur or when Neumann-type boundary conditions are enforced (Zha et al., 2017).

   The PC scheme, on the other hand, corrects the linearization error with an explicit corrector, which avoids iterating and guarantees convergence. The drawback of the PC scheme is that the corrector typically requires smaller time step $\Delta t$ (in comparison to the fully implicit MP scheme), which could potentially increase the computational cost. Moreover, the PC

scheme does not strictly (globally) conserve mass (Lai and Ogden, 2015), although Li et al. (2021) found that the amount of mass lost is often negligible. More detailed description of the PC and MP schemes, as well as a comparison between the two schemes, is available in Li et al. (2021, 2024). It should be noted that using relatively small $\Delta t$ in the PC scheme is not necessarily unacceptable as SERGHEI-RE will be coupled with the SERGHEI-SWE solver in the future. The $\Delta t$ of the SWE solver is restricted by the Courant–Friedrichs–Lewy (CFL) condition for stability. If the Richards solver uses very large $\Delta t$ (i.e.,

the surface and the subsurface modules are asynchronous), the error on the modeled surface-subsurface exchange flux could be non-negligible for rapidly varying surface flow (Li et al., 2023). Thus, to prepare for model coupling and to acknowledge that no numerical scheme is optimal for the Richards equation under all circumstances, both PC and MP schemes are implemented to provide flexibility for SERGHEI-RE to fit in various simulation conditions and requirements.

   Like many existing Richards solvers (e.g. D'Haese et al., 2007; Zha et al., 2019), SERGHEI-RE uses variable $\Delta t$ to improve

computational efficiency. For the PC scheme implemented in SERGHEI-RE, $\Delta t$ is adjusted based on the maximum change of water content in the previous step. For the MP scheme, $\Delta t$ is adjusted based on the number of linearization iterations, similar to the solution in Hydrus (Šimůnek et al., 2013). Acknowledging that both time stepping schemes are somewhat heuristic, an upper limit, $\Delta t_{\max}$, is defined as a user input parameter to avoid unwantedly large $\Delta t$ during the simulation. Detailed mathematical description of the time stepping strategies can be found in Li et al. (2024).

**2.3   User configuration**

SERGHEI-RE provides various options for users to control the simulation. These options can be categorized as domain characteristics and flow characteristics.

   The subsurface domain is defined using a digital elevation model (DEM) that describes the land surface topography, and a height value that describes the thickness of the subsurface domain in the vertical direction. SERGHEI-RE uses structured

Cartesian grids with uniform resolutions in $x$ and $y$ directions, which matches the resolution of the DEM input file. In the vertical ($z$) direction, Richards solvers often require extremely small grid resolution ($\Delta z$) near the ground surface to resolve the infiltration front. To enhance computational efficiency, variable vertical resolution is allowed in SERGHEI-RE, where $\Delta z$



gradually increases from top to bottom. The $\Delta z$ of the $k$th layer from the top (i.e., the land surface) is calculated via

$$\Delta z_k = \Delta z_{\text{base}} \beta^k, \tag{7}$$

where $\Delta z_{\text{base}}$ is the user-provided grid size of the topmost layer, and $\beta \geq 1$ is the ratio of grid size increment.

SERGHEI-RE enables terrain-following mesh to resolve topographic variations in the real world. When complex topography exists, the Darcy fluxes (Eq. 2) in $x$ and $y$ directions are modified correspondingly to conform the terrain slope. The modified Darcy's Law to calculate the horizontal fluxes considering the terrain slope reads (Maxwell, 2013)

$$\boldsymbol{q} = \boldsymbol{K} \left[ cos\omega \nabla \left( h - z \right) + sin\omega \right]. \tag{8}$$

where $\omega$ is the angle between the connection of two cell centers and the horizontal axis. Note that (i) when applying Eq. (8), the bottom boundary of the subsurface domain follows the terrain slope too, and (ii) the vertical flux is unchanged when terrain slope exists.

The subsurface domain can be heterogeneous. To model a heterogeneous system, the user should provide $N$ groups of soil parameters (each group includes the saturated hydraulic conductivity, saturated/residual water content, $\alpha$ and $\tilde{n}$ of the VG

model), and a soil type index (ranging from 0 to $N-1$) for each grid cell. In this way, the heterogeneous subsurface domain can be described on a cell-by-cell basis. Figure 2 shows an example of 2 soil types indexed as 0 and 1. Currently SERGHEI-RE does not yet support multi-porosity subsurface domains.

The flow characteristics consist of the initial and boundary conditions and the source/sink terms. SERGHEI-RE accepts various types of initial and boundary conditions. The user can specify spatially-distributed initial pressure head, water content,

or can specify the initial elevation of the groundwater table. For the latter case, the initial pressure head will be calculated following hydrostatic pressure distribution. Both Dirichlet (constant, space-varying or time-varying pressure head) and Neumann (constant, space-varying or time-varying flux) boundary conditions are accepted for all the boundaries. For lateral boundaries, water table elevation is also an accepted form of boundary condition, which is then converted to hydrostatic pressure distribution along the boundary. For the top boundary, a special interface is implemented to allow reading variables from the SWE

solver, which is reserved for surface-subsurface model coupling in the future. The boundary conditions can be applied to the entire topological boundary surface, or only to a certain subset of it. Similar to SERGHEI-SWE, this is achieved by defining a "polyhedron" that delineates a range in space in which boundary conditions are enforced. The idea of "polyhedron" is also used for defining internal source/sink terms. For example, users can delineate a cuboid in the 3D subsurface domain, within which a flux source (e.g., irrigation) or sink (e.g., root water uptake) term is applied. Similar to the boundary conditions, the

source/sink data can be constant, space-varying or time-varying.

## 3 HPC Implementation

Similar to SERGHEI-SWE, the SERGHEI-RE model features both shared memory and distributed memory parallelism on both multicore CPUs and GPUs (Fig. 1). The performance-portable shared memory parallelism is implemented under the Kokkos





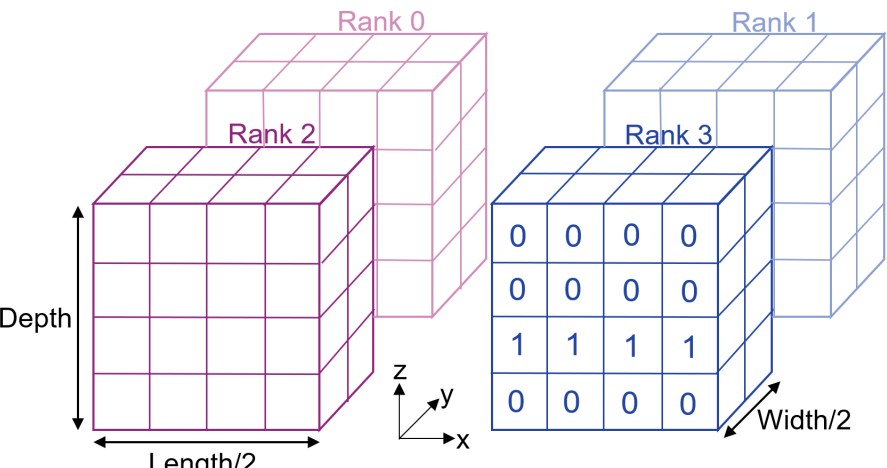

**Figure 2.** An example of SERGHEI-RE domain decomposition with 2 subdomains in $x$ and $y$ directions respectively. The $z$ direction is not decomposed. In Rank 3, the cell-by-cell soil indices are labeled, which consist of 2 types of soil that form 3 layers.

framework, which has been described in detail in Caviedes-Voullième et al. (2023). The distributed memory parallelism is

implemented through MPI. The 3D subsurface domain is partitioned with a 2D domain decomposition in the $x$ and $y$ directions and the subsurface state variables (e.g., $h$, $\theta$) along the boundaries are exchanged between adjacent subdomains. The key difference between the data exchange in SERGHEI-RE and SERGHEI-SWE is that in SERGHEI-RE, the internal subdomain boundaries (i.e., the boundaries of subdomains on which data exchange occurs) are planar rather than linear. As a result, the MPI send and receive functions have been rewritten to manipulate the planar data. To facilitate the coupled surface-subsurface

model development in the future, the vertical coordinate of the subsurface domain is not partitioned into subdomains. This ensures that surface hydrodynamics are modeled in all subdomains, which achieves better load balancing (Fig. 2).

For both the PC and the MP schemes, SERGHEI-RE requires a linear solver to solve for the pressure head from the linearized Richards equation (Li et al., 2024). To complete this task, the preconditioned conjugate gradient (PCG) solver provided in Kokkos Kernels is implemented in SERGHEI-RE. Kokkos Kernels is a linear algebra library under the Kokkos ecosystem.

The main advantage of using Kokkos Kernels is that it is compatible with the Kokkos framework in terms of data structures, computing spaces and overall performance portability. This is demonstrated through the scaling test in Section 5. Finally, it should be pointed out that when the domain is decomposed, each subdomain forms its own linear system. Data transferred across subdomains is inevitably lagged by one time step, but this has negligible influence on solution accuracy for the numerical tests performed in this study.



**Table 1.** A summary of model verification tests. D and N refer to Dirichlet (pressure of water table) and Neumann (flux) boundary conditions (BCs), respectively. Top, bottom, and sides indicates which boundary the BC is applied to. S refers to a source or sink. Soil column indicates whether the soil texture is homogeneous (H) or heterogeneous (NH). Slope column indicates the presence of a topographic slope.

| Section | Dimension | Type of IC | Type of BC | Soil | Slope | Source |
|---------|-----------|------------|------------|------|-------|--------|
| 4.1 | 1D | Dry | D(top) | H | – | Warrick et al. (1985) |
| 4.2 | 1D | Saturated | D(bottom) | H | – | Abeele (1984) |
| 4.3 | 2D | Dry | N(top) | NH | No | Kirkland et al. (1992) |
| 4.4 | 2D | Pressure | D(sides), N(top) | H | Yes | Morway et al. (2013) |
| 4.5 | 2D | Water table | D(top,bottom) | H | Yes | Chávez-Negrete et al. (2018) |
| 4.6 | 2D | Water Content | N(top,bottom)+S | NH | No | He et al. (2018) |

## 4  Model Verification

This section reports the benchmark tests for verifying SERGHEI-RE. Table 1 summarizes the six test problems used herein, including the domain dimension, the types of initial/boundary condition, soil heterogeneity and terrain slope. In addition to traditional hydrological tests, we also include tests in the field of geotechnical engineering (Section 4.5) and agriculture (Section 4.6) to demonstrate the broad range of potential applicability of SERGHEI-RE. Results of the first two test problems are generated on an Intel Core i9-9880H processor (8 cores, 16 threads, 2.3GHz). The other four tests are completed on an Nvidia RTX A5000 GPU (24GB memory, 768GB/s bandwidth).

### 4.1  1D infiltration

The first test is a 1D infiltration problem from Warrick et al. (1985). A fixed pressure head of 0m is applied to the top of a soil column. The column is nearly dry, with a uniform water content of 0.03 as the initial condition. This problem has been widely used to validate Richards solvers, because analytical solutions have been derived (Caviedes-Voullième et al., 2013; Lai and Ogden, 2015; Li et al., 2021; Phoon et al., 2007). A total of eight simulation scenarios are established, with two different numerical schemes (PC, MP), two different $\Delta t_{\max}$ (0.4s, 10s), and three different grid resolutions: $\Delta z$ =0.01m, $\Delta z$=0.02m and variable $\Delta z$ between 0.01m (top) and 0.03m (bottom). The three grid resolutions result in 100, 50 and 56 grid cells respectively along the soil column.

Figure 3 shows the water content profiles modeled with SERGHEI-RE (all with $\Delta t_{\max}$=0.4s), as well as the analytical solutions at the corresponding times. It can be seen from Fig. 3(a) that with $\Delta z$ =0.01m, both the PC and the MP schemes achieve good agreements with the analytical solution. As $\Delta z$ increases, the infiltration speed is overestimated. Clearly, the solution accuracy is sensitive to the grid resolution (Li et al., 2021). A closer look at the infiltration fronts (Fig. 3b) reveals negligible difference between the PC and the MP schemes.

More insights are obtained from Table 2, where the root mean square errors (RMSE, in terms of the depth difference between the modeled infiltration fronts and the analytical solution) and the computational cost are listed. Compared with a uniform $\Delta z$



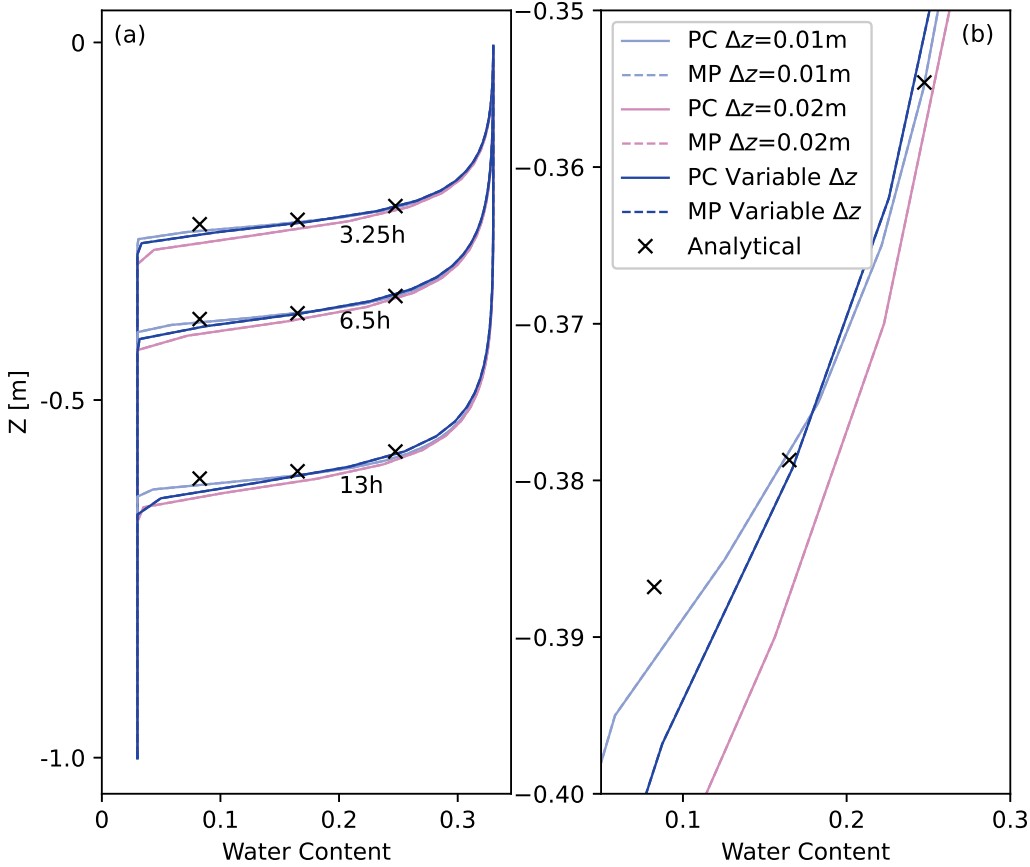

**Figure 3.** SERGHEI-RE simulation results (water content profiles) of the Warrick's problem against the analytical solution with different choices of $\Delta z$. (a) shows the water content profiles over the entire column, and (b) zooms in to examine the profiles in detail.

of 0.02m, using variable $\Delta z$ results in approximately 40% lower RMSE, with only about 7% increased computational cost. With the same $\Delta t_{\max}$, the MP scheme is computationally more expensive than the PC scheme, because it is iterative. However, when $\Delta t_{\max}$ is set to 10s, the RMSE of the MP scheme remains low, but the RMSE of the PC scheme increases by 25%. This indicates the error of the PC scheme is more sensitive to $\Delta t_{\max}$, because the corrector step is fully explicit (Li et al., 2021, 2024). As a result, the MP scheme with $\Delta t_{\max}$=10s is much faster than the PC scheme with $\Delta t_{\max}$=0.4s, despite their comparable RMSEs.

### 4.2 1D drainage

The second test is based on a free drainage experiment of Abeele (1984), which has been previously used for model verification (Caviedes-Voullième et al., 2013; Forsyth et al., 1995; Lai and Ogden, 2015). The computational domain is a homogeneous,



**Table 2.** A summary of the 8 simulation scenarios of the 1D infiltration problem. The numerical scheme, $\Delta z$, $\Delta t_{\max}$, RMSE and the simulation time are listed.

| Numerical Scheme | $\Delta z$ [m] | Number of Cells | $\Delta t_{\max}$ [s] | RMSE [cm] | Runtime [s] |
| --- | --- | --- | --- | --- | --- |
| PC | 0.01 | 100 | 0.4 | 0.5823 | 161.6 |
| MP | 0.01 | 100 | 0.4 | 0.5811 | 175.8 |
| PC | 0.02 | 50 | 0.4 | 1.6483 | 98.4 |
| MP | 0.02 | 50 | 0.4 | 1.6484 | 108.6 |
| PC | Variable | 56 | 0.4 | 0.9339 | 104.5 |
| MP | Variable | 56 | 0.4 | 0.9329 | 115.7 |
| PC | 0.01 | 100 | 10 | 0.7308 | 10.9 |
| MP | 0.01 | 100 | 10 | 0.5987 | 35.1 |

initially fully saturated 1D soil column. The bottom of the column is open with a fixed pressure equal to atmospheric pressure ($h$ =0m). The simulation period is set to 100 days, with $\Delta z$=0.06m and $\Delta t_{\max}$=1 day for both PC and MP schemes. The modeled water content profiles can be found in Fig. 4. SERGHEI-RE generally reaches good agreements with the experimental results, with slight underestimation of the water content on day 100. The PC and MP schemes exhibit negligible difference on
this test problem, but the MP scheme requires almost 3 times the computational cost (3.83s versus 1.33s).

### 4.3   2D infiltration with layered soil

The third test problem simulates infiltration into layered soil, which has been used for model verification in Bassetto et al. (2022); Kirkland et al. (1992); Forsyth et al. (1995). The model domain (Fig. 5) is 2D rectangular, and consists of two types of soils. Initially, the domain is nearly dry with a uniform pressure head of -400m. A constant infiltration flux is applied to the
middle section of the top layer. Since *Soil2* has lower permeability than *Soil1*, as the wetting front infiltrates downward, water will accumulate near the interface between the two soil layers. For this test problem, the MP scheme shows a slow convergence rate. Thus, only the PC scheme is used (with $\Delta z$=0.1m and $\Delta t_{\max}$=10s) to generate results for model verification. As can be seen from Fig. 6, a saturated zone characterized by a positive pressure head is formed at the soil interface as a result of the permeability change. This trend is well captured with SERGHEI-RE.

### 4.4   2D infiltration with open lateral boundaries

The fourth problem models rainfall onto an inclined plane with open lateral boundaries (Beegum et al., 2018; Brandhorst et al., 2021; Morway et al., 2013). An illustration of the problem setup is shown in Fig. 7. The water table depth is fixed at 8m and 14.1m on the left and right boundaries, respectively. Initially, hydrostatic pressure distribution is applied below the water table and a constant pressure head is enforced above the water table. Rainfall with variable intensity is applied to the top boundary (Table 3). The original problem is 3D. However, since all the grid layers in the $y$ direction are identical, for model verification



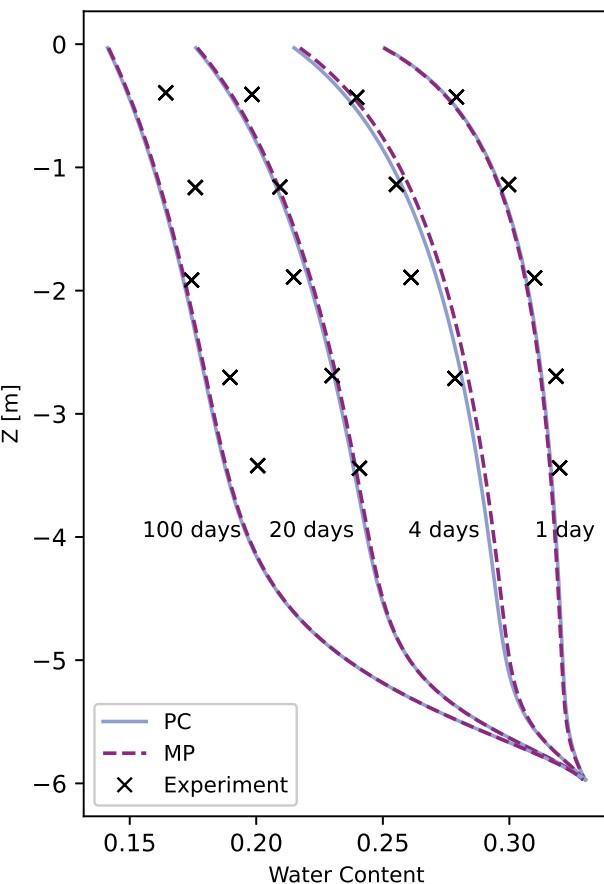

**Figure 4.** SERGHEI-RE simulation results (water content profile) of the drainage problem against the experimental data. The simulations use $\Delta z ==0.06$m and $\Delta t_{max}=1$ day.

**Table 3.** Monthly variable rainfall intensity used for the 2D infiltration (with lateral flow) test problem. The data shown is for one year. It is assumed that each year in the simulation has the same rainfall intensity.

| Month | 1 | 2 | 3 | 4 | 5 | 6 |
|---|---|---|---|---|---|---|
| Rainfall [mm/h] | 0.017 | 0.017 | 0.042 | 0.071 | 0.1 | 0.079 |
| Month | 7 | 8 | 9 | 10 | 11 | 12 |
| Rainfall [mm/h] | 0.108 | 0.088 | 0.054 | 0.046 | 0.025 | 0.017 |

purposes, a 2D $x$-$z$ simulation is sufficient. The 3D version of this test problem is used for the scaling tests described in Section 5.



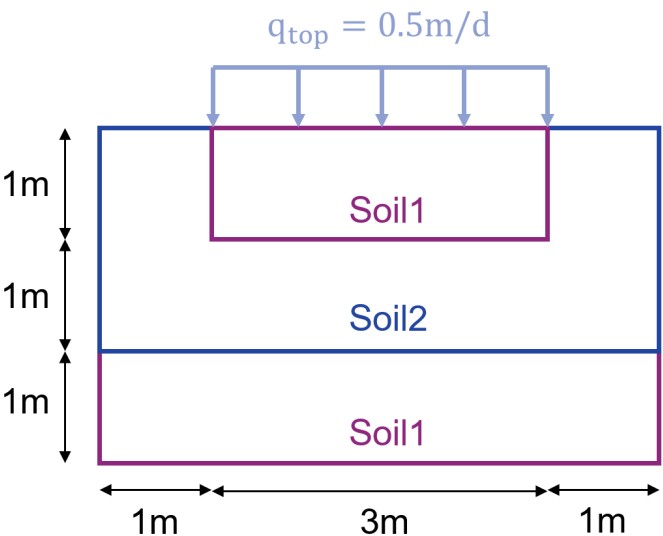

**Figure 5.** Sketch of the model domain for the heterogeneous infiltration problem.

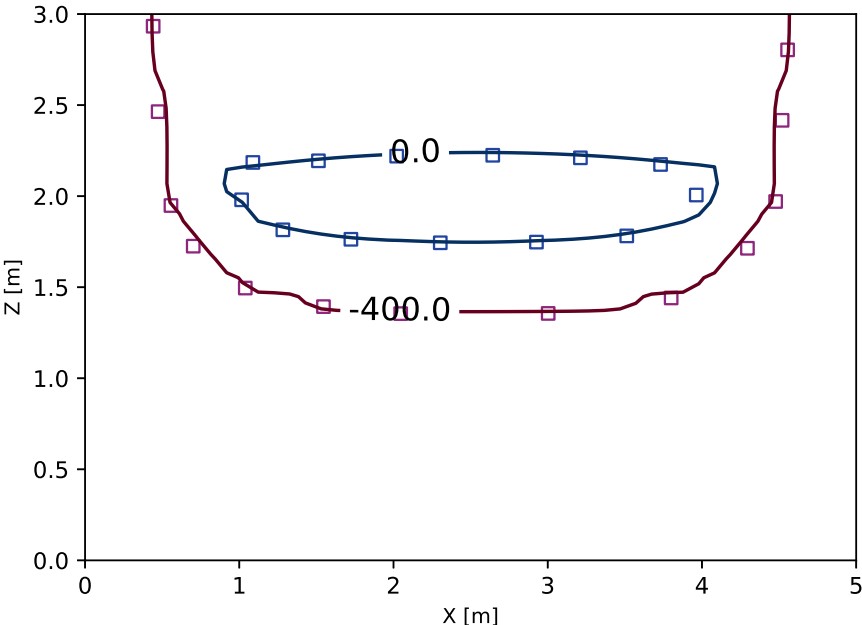

**Figure 6.** The pressure head contours modeled with SERGHEI-RE (solid lines, with PC scheme) and the reference solution (square markers, which is the numerical solution reported in Kirkland et al. (1992)) for the 2D infiltration problem at the end of the simulation (1 day). The numbers on the contours are the pressure head values in meters.



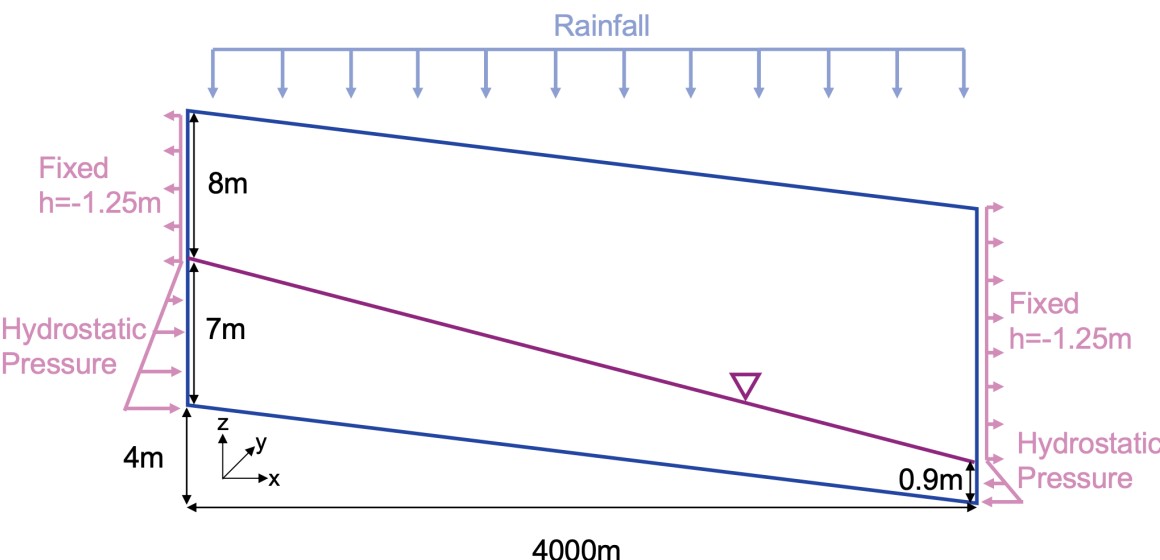

**Figure 7.** The dimension of the model domain and the boundary conditions for the 2D infiltration problem with open lateral boundaries.

**Table 4.** A summary of the 4 simulation scenarios of the 2D infiltration (with lateral flow) problem. The numerical scheme, $\Delta z$, and the simulation time are listed.

| Numerical Scheme | $\Delta z$ [m] | Number of Cells | Runtime [min] |
|:---:|:---:|:---:|:---:|
| PC | 0.25 | 2400 | 22.45 |
| MP | 0.25 | 2400 | 38.65 |
| PC | 0.1-0.81 | 1760 | 13.34 |
| MP | 0.1-0.81 | 1760 | 25.49 |

Four simulation scenarios are established for this test case: uniform grid resolution ($\Delta z =0.25$m) with the PC and the MP schemes, and variable grid resolution ($\Delta z_{\text{base}} =0.1$m, $\beta =1.05$) with the PC and the MP schemes. The uniform and variable resolution scenarios have 60 and 44 grid cells in the vertical direction respectively. All the simulations are performed for five years with $\Delta t_{\text{max}}=3600$s. It can be seen from Table 4 that by applying variable grid resolution the total wall-clock simulation time reduces by 41% and 34% for the PC and the MP schemes, respectively. With the same $\Delta t_{\text{max}}$ the MP scheme is more expensive than the PC scheme. These trends are consistent with those reported in Section 4.1 and 4.2.

When rainfall penetrates the top boundary, driven by the horizontal pressure gradient, the infiltrated water will exit the domain through its lateral boundaries. This results in a concave water table. Furthermore, since the rainfall rate varies monthly, the water table elevation fluctuates in time. Figure 8(a) shows the temporal variation of the water table elevation in the center of the domain and Fig. 8(b) shows the water table elevation at the end of the five-year simulation. It can be seen that all



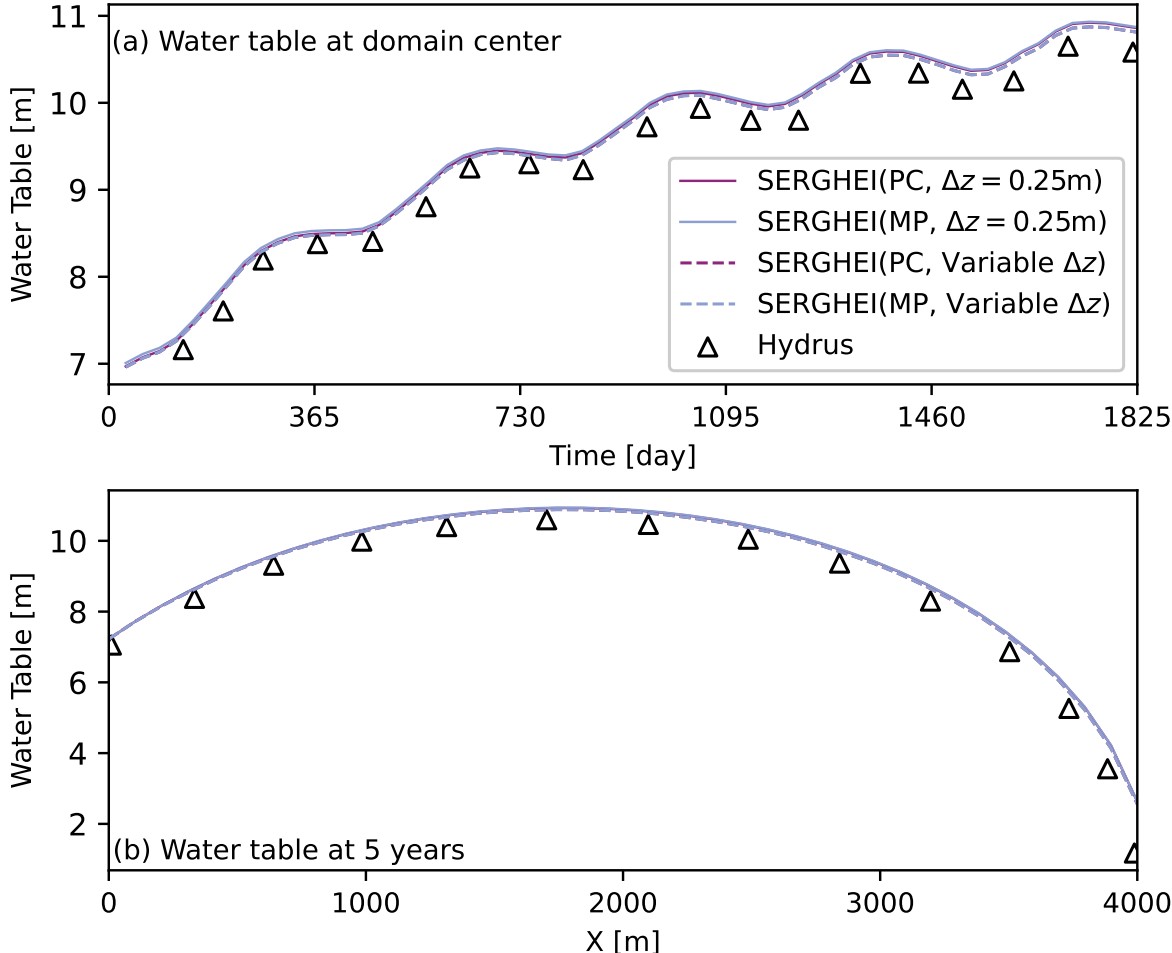

**Figure 8.** SERGHEI-RE simulation results (water table elevation) of the infiltration problem against the Hydrus results.

SERGHEI-RE scenarios have good agreements with Hydrus2D, which is used as the reference solution herein. Differences in-between SERGHEI-RE scenarios are negligible for this test problem.

## 4.5 2D infiltration into a road embankment


The fifth problem models water infiltration into a 2D partially-paved road embankment (Fig. 9). In the original problem, the bottom of the embankment is flat with a constant pressure head of 0m (Chávez-Negrete et al., 2018). For SERGHEI-RE, however, the terrain-following mesh requires an extended bottom section resulting in an inclined bottom boundary. To preserve the original water table elevation, a hydrostatic pressure head (calculated from the original water table elevation) is enforced



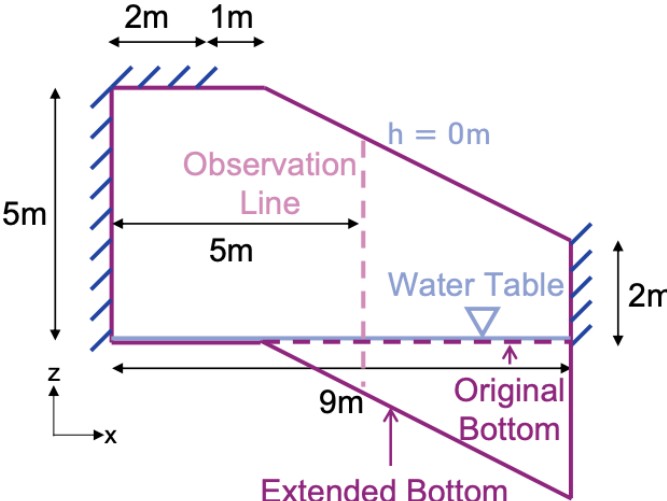

**Figure 9.** Sketch of the model domain and the boundary conditions for the road embankment problem. Dashed lines indicate the required extension due to the terrain following mesh.

along the inclined bottom boundary. A thin layer of water ($h$=0m) is applied to the open top boundary. The simulation is performed with the PC scheme only. The MP scheme is difficult to converge for this test problem.

Figure 10 shows a comparison between the pressure profiles at the observation line (Fig. 9) modeled with SERGHEI-RE and with the generalized finite difference method (GFDM) of Chávez-Negrete et al. (2018). The two results achieve good agreement up to day 1. At 1.5 days, SERGHEI-RE slightly overestimates the bottom pressure compared to GFDM. This is due to the reason that for the SERGHEI-RE domain, the pressure boundary condition is enforced along the inclined bottom, not the location of the water table. Thus, during the simulation, the actual water table might slightly deviate from $z$=0m.

### 4.6 2D crop irrigation

The sixth problem models soil water dynamics in a real agricultural field. Prior field measurements and numerical simulations at this site are documented in He et al. (2018). The model domain is a 2D vertical transect of the crop field with dimensions, soil layers, and boundary conditions illustrated in Fig. 11(a). Note that the model domain built herein is half of the original domain, because the original domain is fully symmetric. The ground surface is partially covered with film mulch, below which drip irrigation is applied. The side boundaries are impermeable and the bottom boundary is free drainage. The rainfall, evaporation, irrigation, and transpiration rates applied are shown in Fig. 11(b) and (c) for year 2014 and 2015, respectively. The grid resolutions are $\Delta x$ =0.01m and $\Delta z$=0.01m. The maximum $\Delta t$ is set to 60s for both PC and MP schemes tested.

It should be noted that SERGHEI-RE is not particularly designed for agricultural applications. For the time being, SERGHEI-RE does not consider spatially variable and pressure dependent root water uptake, which is different from the treatment in most agricultural hydrologic models (including He et al. (2018)). To reproduce the field measurements and numerical modeling





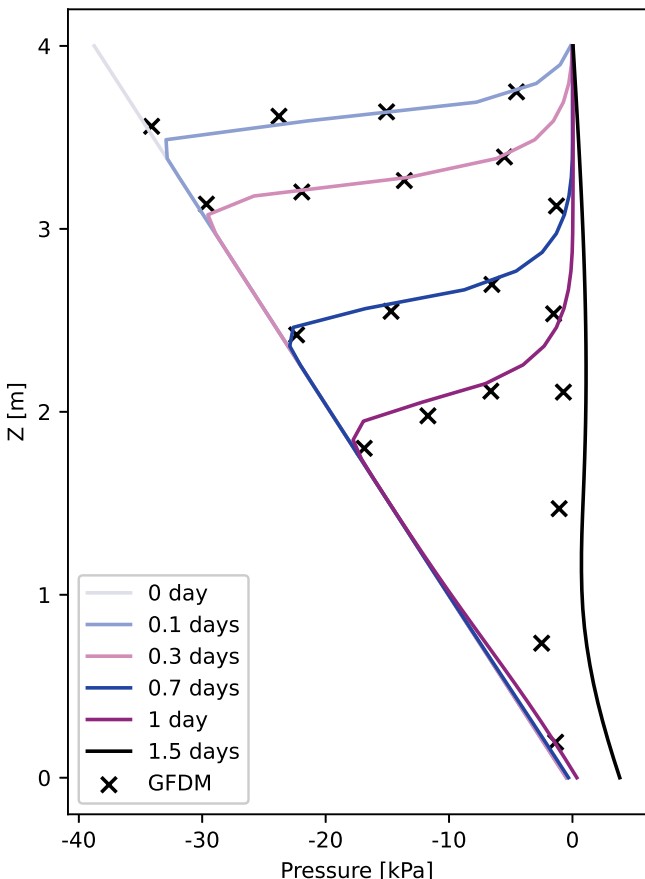

**Figure 10.** SERGHEI-RE simulation results of the infiltration problem against the GFDM results.

results of He et al. (2018), the transpiration rate is treated as a sink term that is uniformly applied on the root zone, whose dimension is used to calibrate SERGHEI-RE. The final width and depth of the root zone is set to be 0.5m and 0.4m, respectively.

Although these features of the problem cannot be fully represented in SERGHEI-RE, this problem is interesting for its multiple boundary fluxes (both inflow and outflow) as well as internal source/sink terms. It is therefore useful to test and validate such features in SERGHEI-RE.

Figure 12 shows the evolution of water content at 0.2m and 0.6m beneath the irrigation emitter. For year 2014 and 2015, the agreement between SERGHEI-RE and Hydrus2D is reasonably good. The irrigation events, which are implied by the rising

water content, are well captured by SERGHEI-RE. The PC and MP schemes produce indistinguishable results on this test problem.

The computational domain consists of four soil layers, four types of boundary conditions (rainfall, evaporation, irrigation, and free drainage), and one internal sink term (root water uptake) making it a challenging test case from a modeling perspective.



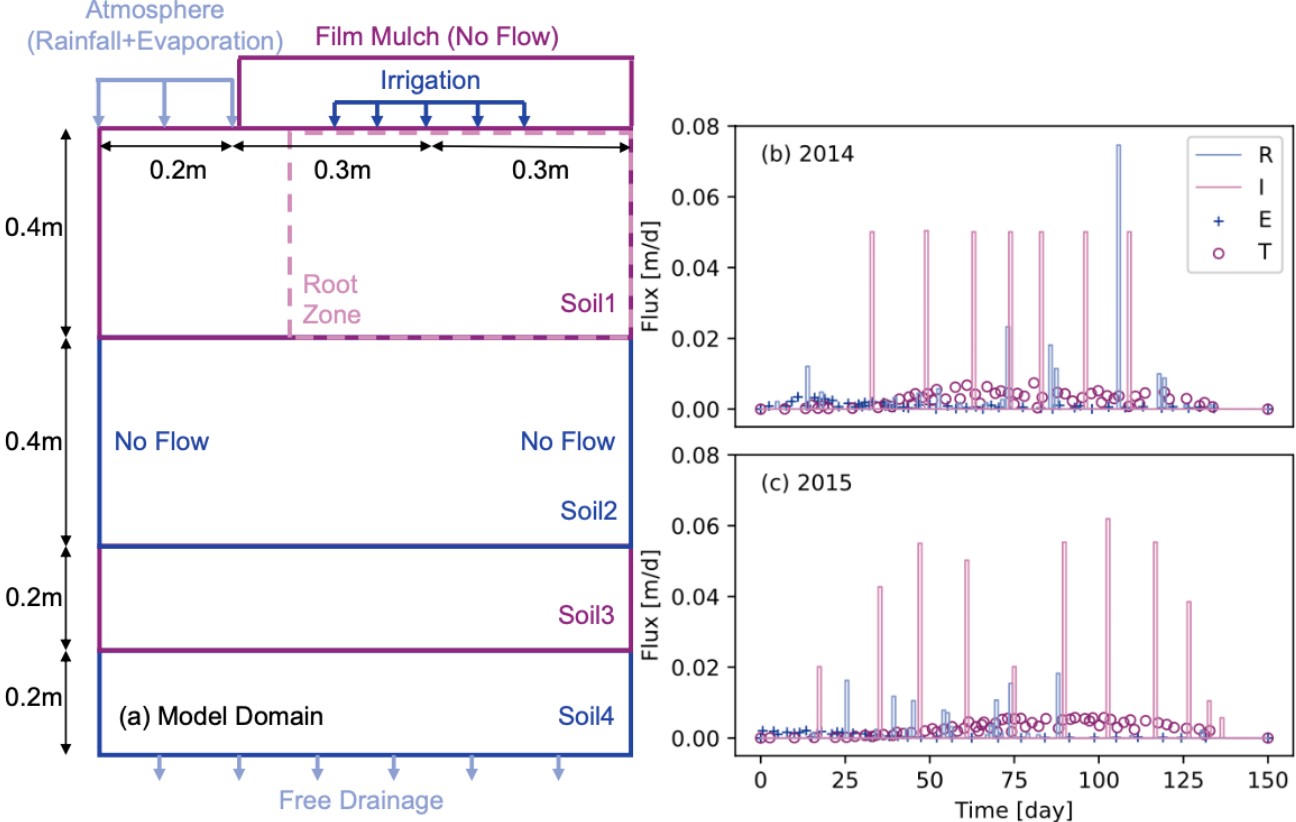

**Figure 11.** (a) The dimension of the model domain and the boundary conditions for the crop irrigation problem, (b) the rainfall (R), irrigation (I), soil evaporation (E) and root water uptake (transpiration, T) rates during the study period in 2014, (c) R, I, E and T during the study period in 2015.

Figure 12 demonstrates the capability of SERGHEI-RE in simulating soil water dynamics under complex field conditions.
Model discrepancies can be attributed to (i) the uniform root water uptake rate used in SERGHEI-RE, and (ii) the possible error in the input data. For example, the first irrigation event in 2015, as evidenced by the rising water content modeled with Hydrus2D (Fig. 12c), is not documented in the irrigation data (He et al., 2018). This might directly lead to the underestimation of water content in the first 40 days of Fig. 12(c) and (d).

## 5 Performance and Scaling

An important feature of the SERGHEI-RE model is scalability and performance portability. SERGHEI-RE is designed both for simple, idealized simulations on personal laptops and desktops, and for large-scale, realistic simulations on HPCs. To demonstrate the scalability and performance portability of SERGHEI-RE, the rainfall infiltration problem (Section 4.4) is



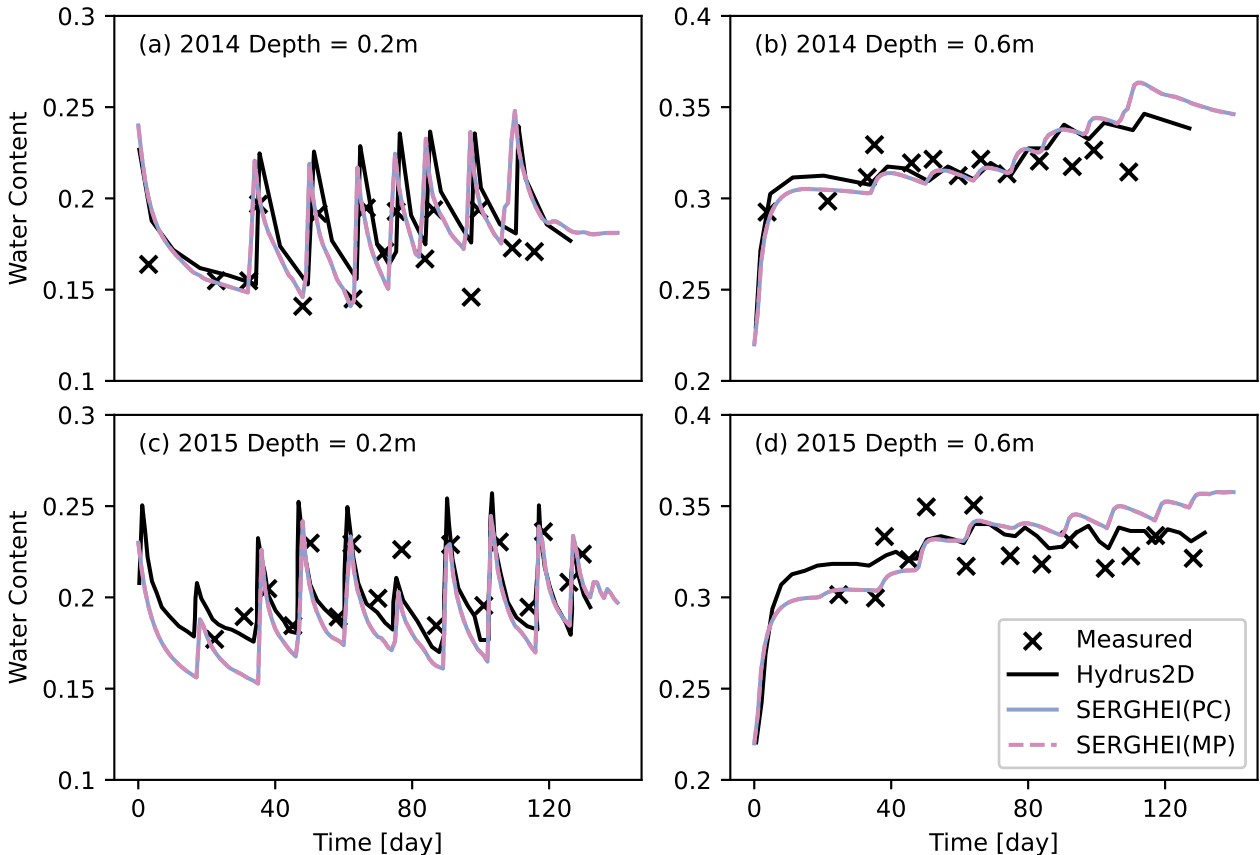

**Figure 12.** SERGHEI-RE simulation results (water content at 0.2m and 0.6m beneath the irrigation dripper) of the crop irrigation problem against field measurements and Hydrus2D results.

extended to 3D with finer grid resolutions ($\Delta x$, $\Delta y$ = 5m, $\Delta z$ = 0.1m). By adjusting the width of the computational domain in $y$ direction, rainfall infiltration simulations with different numbers of grid cells can be created—120 000 cells for each additional row in y direction. This way, the need for varying domain sizes regarding scaling tests on different computational platforms can be accomplished. All other model inputs, such as the soil properties, the initial and boundary conditions, are kept same as in Section 4.4. We demonstrate the scaling test results on three computational domains: a small domain with 1.2 million grid cells (10 rows in y direction), a medium domain with 15.36 million grid cells (128 rows), and a large domain with 120 million grid cells (1 000 rows).



## 5.1 Small domain with 1.2 million cells

For the small domain test case, SERGHEI-RE is run with both the PC and the MP schemes on two computational systems: (i) a Dell Precision 5820 desktop workstation equipped with an Intel-Xeon W-2265 CPU (3.5 GHz, 12 cores, 24 threads) and an Nvidia RTX A5000 GPU (24 GB memory, 768 GB/s bandwidth, 8192 CUDA cores), and (ii) a small cluster at the high-performance computing center of Tongji University (named TJ HPC hereafter), where each CPU node is equipped with an Intel Xeon Max 9468 processor (3.5 GHz, 48 cores, 96 threads), and each GPU node is equipped with an Nvidia L40 GPU (48 GB memory, 864 GB/s bandwidth, 18176 CUDA cores). For both systems, shared memory parallelization (OpenMP) is tested on a single CPU using a various number of threads. On TJ HPC, an additional scenario is tested, where the computation task is split into 4 MPI processes on 4 nodes. Each node uses 1 to 64 OpenMP threads, beyond which the maximum allowed computational resources on TJ HPC is met. GPU tests are completed on single A5000 and L40 GPUs respectively. Thus, systematic multi-CPU or multi-GPU parallelization tests are not performed herein—they are reserved for the medium and the large domains. For this test, the simulation length is set to two hours with $\Delta t_{\max}$=180s.

Figure 13 shows the simulation time and the speedup for different parallel configurations. It can be seen that on the desktop CPU, the scaling is nearly ideal up to 12 threads. With 16 threads, the scaling begins to deteriorate. On the TJ HPC, the speedup gradually deviates from linear beyond 16 threads, but keeps increasing even at full capacity (96 threads). For the 4 nodes scenario, the simulation scales well from 16 to 192 threads, then deteriorates at 256 threads. As expected, with the same $\Delta t_{\max}$, the MP scheme is computationally more expensive than the PC scheme, but the speedups are similar. This demonstrates, with the same amount of total threads, splitting the work into multiple nodes effectively improves scaling. With a single GPU, the simulation time is dramatically reduced. The speedups are 89.4 and 269.7 relative to a single CPU thread for the A5000 and L40 GPUs, respectively. If compared to 96 threads of TJ-HPC, the speedups are 1.7 and 6.1, respectively. This indicates that even on desktop workstations, GPU-based SERGHEI-RE significantly enhances the computational efficiency compared to traditional, CPU-based Richards solvers, making it a promising tool for large-scale variable-saturated subsurface flow simulations.

## 5.2 Medium domain with 15.36 million cells

In this test case, SERGHEI-RE is deployed on the LISE HPC system of the German National High Performance Computing (NHR) Alliance at Zuse Institute Berlin, Germany. Each computation node of LISE is equipped with two units of Intel Xeon Platinum 9242 processors (2.30 GHz, 48 cores, 96 threads). The simulation length is set to 6 hours with $\Delta t_{\max}$=180s. The simulation time and the speedup (with the PC scheme) for different parallel configurations is shown in Figure 14. The speedup curve is approximately 73-84% efficient compared to ideal linear scaling from 2 to 128 nodes with the observed maximum efficiency of 84.4% at 32 nodes. Increasing deterioration in scaling emerges beyond 128 nodes with 256 and 512 nodes being 51% and 27% efficient, respectively. These results indicate that SERGHEI-RE achieves good scaling performance up to 128 nodes for the medium domain on CPU-based HPC systems, successfully demonstrating its performance portability. Table 5 shows that the PCG solver of Kokkos Kernels takes more than 80% of the total simulation time from 1 to 128 nodes and decreases with more nodes being used. Furthermore, the speedup of the linear solver is slightly higher than the overall speedup



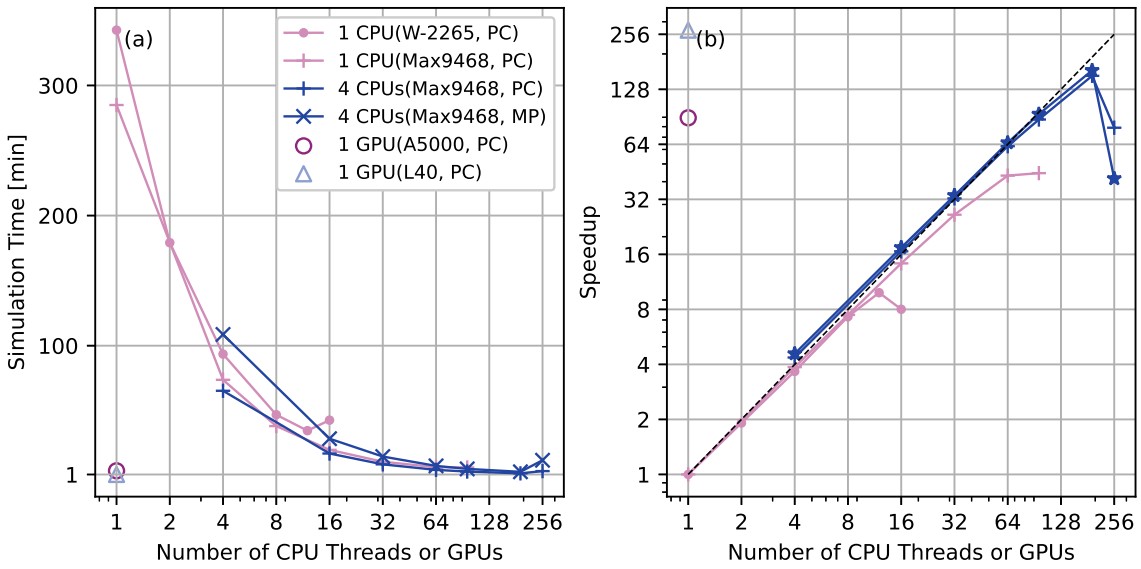

**Figure 13.** (a) SERGHEI-RE simulation time and (b) speedup on a desktop workstation and on the TJ HPC. The test problem has 1.2 million grid cells. Note that the horizontal axis is the number of CPU threads (i.e., the number of CPUs times the number of threads used per CPU) or the number of GPUs.

of the simulation, suggesting the scaling performance is mainly attributed to the PCG solver. A similar finding is reported in Li et al. (2024).

### 5.3 Large domain with 120 million cells

For testing the large domain, SERGHEI-RE is deployed on the JUWELS (cluster and booster) system at the Jülich Super-computing Center. Each JUWELS cluster GPU node contains 4 Nvidia Tesla V100 GPU cards (32 GB memory, 900 GB/s bandwidth). Each JUWELS booster node contains 4 Nvidia A100 Tensor Core GPU cards (80 GB memory, 1935 GB/s bandwidth). The simulation length is set to 6 hours with $\Delta t_{\max}$=180s. Figure 15 shows the simulation time and the speedup on JUWELS cluster and JUWELS booster nodes. It can be seen from Fig. 15 that up to 64 nodes the speedup curves are approximately linear on both V100 and A100 GPUs, and for both PC and MP schemes. Superlinear speedup is observed from 2 to 32 GPU nodes. Beyond 64 GPU nodes the scaling starts to deteriorate with the PC scheme showing stronger deterioration than the MP scheme. This is because the linear system solver takes a greater portion of the total simulation time for the iterative MP scheme, and the linear solver scales well (see Li et al. (2024) and Table 5 for further discussion). Figure 15 clearly shows that the MP scheme is substantially more expensive than the PC scheme, although they use the same $\Delta t_{\max}$. Nonetheless, the scaling behaviour is very similar with fewer than 64 nodes, which is a clear indication that the number of linearization steps in the MP is responsible for the computational overhead.



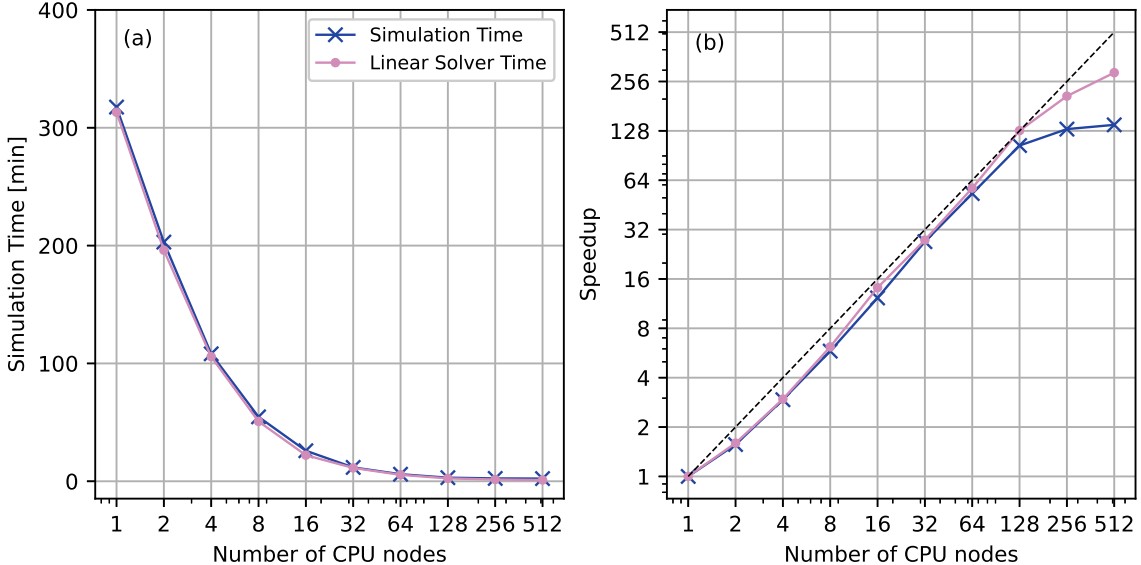

**Figure 14.** SERGHEI-RE simulation time and speedup on the LISE HPC with Intel Xeon Platinum 9242 CPUs. The test problem has 15.36 million grid cells.

Table 5 illustrates that the linear system solver (i.e., the PCG solver of Kokkos Kernels) takes more than 94% of the total simulation time for 1 to 32 GPU nodes. Similar to LISE, with more nodes used, the proportion of solver time decreases on JUWELS booster. Furthermore, the speedup of the linear solver is higher than the overall speedup as more nodes are used. Again, this means that the good scaling performance is mainly attributed to the good scaling of the Kokkos Kernels solver. At 160 nodes, the linear solver only takes 61.7% of the simulation time. Although the solver speedup exceeds 120, the overall speedup is only 78.93. Detailed examination (not shown) reveals that at 160 nodes, the MPI exchange time is comparable with the linear solver time, indicating that given the dimension of this test problem, too many GPUs are used, which squeezes the dimension of the subdomains, resulting in sub-optimal scaling performance (with 160 nodes, each subdomain contains $25 \times 50 \times 150$ grid cells in x, y, and z directions, respectively). Table 5 also provides the ratio of simulation times between the MP and PC schemes on the JUWELS booster, which is always greater than one. This result is consistent with Fig. 15, illustrating that given the same $\Delta t_{\max}$, the MP scheme is computationally more expensive than the PC scheme regardless of the parallelization configurations. Overall, Table 5 demonstrates similar scaling performance (i.e., near-linear to superlinear speedup, good scaling of the linear solver) on both multi-CPU and multi-GPU HPC systems, highlighting the performance portability of the SERGHEI-RE model.



**Table 5.** Speedup comparison for the entire simulation, the linear solver (PCG) as well as the proportion of the linear solver on the runtime for the PC scheme scenarios on the LISE HPC (Fig. 14) and the JUWELS booster HPC (Fig. 15). Efficiency is determined as the quotient of speedup and node count. The last row shows the ratio of simulation times between MP and PC schemes on the JUWELS booster.

| LISE Nodes | 1 | 2 | 4 | 8 | 16 | 32 | 64 | 128 | 256 | 512 |
|---|---|---|---|---|---|---|---|---|---|---|
| Speedup | 1.000 | 1.565 | 2.936 | 5.811 | 12.257 | 27.005 | 53.062 | 104.174 | 131.212 | 139.256 |
| Speedup (PCG) | 1.000 | 1.597 | 2.961 | 6.168 | 14.159 | 27.627 | 57.286 | 128.429 | 208.469 | 289.725 |
| Efficiency | 100.0% | 78.2% | 73.4% | 72.6% | 76.6% | 84.4% | 82.9% | 81.4% | 51.3% | 27.2% |
| Efficiency (PCG) | 100.0% | 79.8% | 74.0% | 77.1% | 88.5% | 86.3% | 89.5% | 100.3% | 81.4% | 56.6% |
| Proportion of runtime | 98.6% | 96.6% | 97.8% | 92.9% | 85.4% | 96.4% | 91.3% | 80.0% | 62.1% | 47.4% |
| JUWELS A100 Nodes | 1 | 2 | 4 | 8 | 16 | 32 | 64 | 160 | 320 | - |
| Speedup | 1.000 | 2.020 | 4.141 | 8.484 | 16.997 | 32.292 | 57.550 | 78.93 | 74.10 | - |
| Speedup (PCG) | 1.000 | 2.022 | 4.159 | 8.574 | 17.446 | 33.928 | 63.979 | 127.67 | 202.06 | - |
| Efficiency | 100.0% | 101.0% | 103.5% | 106.1% | 106.2% | 100.9% | 89.9% | 49.3% | 23.2% | - |
| Efficiency (PCG) | 100.0% | 101.1% | 104.0% | 107.2% | 109.0% | 106.0% | 100.0% | 79.8% | 63.1% | - |
| Proportion of runtime | 99.7% | 99.6% | 99.3% | 98.7% | 97.2% | 94.9% | 89.7% | 61.7% | 36.6% | - |
| Sim. Time (MP/PC) | 3.212 | 3.270 | 3.237 | 3.263 | 3.238 | 3.244 | 3.050 | 2.312 | 1.793 | - |

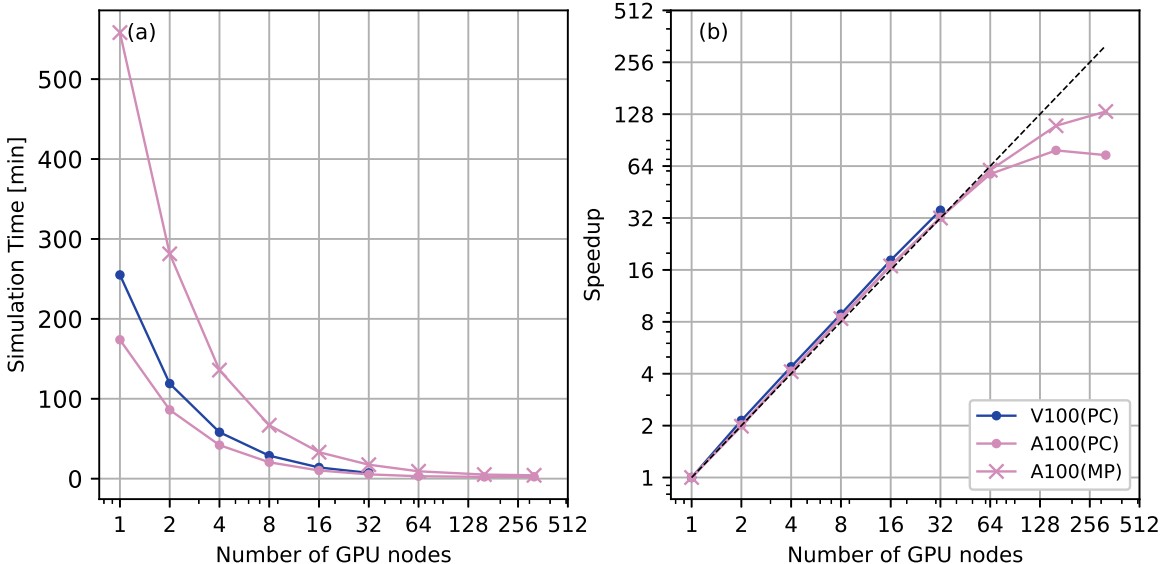

**Figure 15.** (a) SERGHEI-RE simulation time and (b) speedup on Nvidia V100 and A100 GPU equipped on the JUWELS HPC. The test problem has 120 million grid cells.



# 6   Conclusions

In this manuscript, we present SERGHEI-RE, the variably-saturated subsurface flow simulation module of SERGHEI. SERGHEI-RE provides both an iterative and a non-iterative numerical scheme for subsurface flow simulation. This enhances model flex-ibility under various simulation conditions that could make numerical solution to the 3D Richards equation challenging. By testing SERGHEI-RE on six benchmark problems, ranging from simple infiltration and drainage, to realistic hydrological, geotechnical and agricultural applications, we show that SERGHEI-RE produces satisfactory agreements with experimental measurements, field observations and/or simulation results with other softwares. We demonstrate that the numerical scheme of SERGHEI-RE is accurate, robust and versatile.

SERGHEI-RE is developed within the SERGHEI framework, meaning that it is equipped with the Kokkos programming model to achieve performance-portable parallelization on various computational devices. We show that SERGHEI-RE scales well on personal desktop workstations, as well as on multi-CPU and multi-GPU supercomputers, thereby demonstrating its performance-portable scalability that is critical for subsurface flow models aimed towards environmental applications, while simultaneously fully aligning with the evolving computational advancements into the exascale era. SERGHEI-RE shows very good scalability with both the predictor-corrector (PC) and the well-established fully implicit modified Picard (MP) schemes. The MP scheme typically allows for a greater time step size to enhance computational efficiency, but it is computationally more expensive per time step because it is iterative.

Variably-saturated subsurface flow simulation is at the core of hydrological simulations. However, realistic engineering applications also require additional modeling capabilities such as surface-subsurface flow exchange, reactive transport, and vegetation dynamics. As shown in Fig. 1, these (and more) functions are being developed and tested. Additional modules are expected to be released in the near future, extending the applicability of SERGHEI in the broad range of computational hydrology.

*Code and data availability.*

SERGHEI is available through GitLab, at https://gitlab.com/serghei-model/serghei, under a 3-clause BSD license. The SERGHEI-RE source code is tagged as SERGHEI v2.0 (last access: 4 August 2023). A static version of SERGHEI-RE (SERGHEI v2.0) is archived in Zenodo, with DOI: https://doi.org/10.5281/zenodo.13166466 (Caviedes-Voullième et al., 2024). Developer and user guides are available in the wiki page of the SERGHEI project. The following tools and packages are pre-requisite for SERGHEI-RE: GCC (other C++ compilers have not been tested), OpenMPI (other MPI implementations have not been thoroughly tested), CMake, Kokkos, KokkosKernels, PnetCDF.

The input files of the test cases reported in this manuscript are available in the serghei-tests repository on GitLab (https://gitlab.com/serghei-model/serghei-tests/subsurface/analytical and https://gitlab.com/serghei-model/serghei-tests/subsurface/experimental). A static version of the tests is archived in Zenodo, with DOI: https://doi.org/10.5281/zenodo.13282882 (Li, 2024).




*Author contributions.*

ZL contributed to conceptualization, methodology, software, formal analysis, and writing(original draft). GR contributed
to formal analysis and writing(original draft). NZ contributed to software, formal analysis and writing(review & editing). ZZ
contributed to formal analysis and writing(review & editing). IOX contributed to formal analysis and writing(review & editing).
DCV contributed to conceptualization, software and writing(review & editing).

*Competing interests.*

The authors declare no competing interests.

*Acknowledgements.*   This study is supported by the National Natural Science Foundation of China (NSFC Grant No. 42307078), the National
Key R&D Program of China (2022YFC3803000), and the Fundamental Research Funds for the Central Universities (China). IOX and GR
are funded by the *Tenure-Track-Programm* of the German Ministry of Education and Research. This work used HPC resources provided by
the Center for Scientific Computing at Tongji University, China, and the German National High Performance Computing (NHR) Alliance
at Zuse Institute Berlin (ZIB), Germany. The authors gratefully acknowledge the Earth System Modelling Project (ESM) for supporting this
work by providing computing time on the ESM partition of the JUWELS supercomputer at the Jülich Supercomputing Centre (JSC) through
the compute time project *Runoff Generation and Surface Hydrodynamics across Scales with the SERGHEI model* (RUGSHAS), Project
Numbers 22686 and 29000.





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
