# Peer review of "SERGHEI v2.0: introducing a performance-portable, high-performance three-dimensional variably-saturated subsurface flow solver (SERGHEI-RE)"

_EGUsphere, 2024_

## Author Comment (AC1)

**Response to Referee 1 Comments on:**

**SERGHEI v2.0: introducing a performance-portable, high-performance three-dimensional variably-saturated subsurface flow solver (SERGHEI-RE)**

Zhi Li, Gregor Rickert, Na Zheng, Zhibo Zhang, Ilhan Özgen-Xian, and Daniel Caviedes-Voullième

Thank you for your thorough review of our manuscript. We appreciate your constructive feedback and we have addressed your comments point by point in our revision.

**Referee #1 (RC1):**

In the manuscript titled "SERGHEI v2.0: introducing a performance-portable, high-performance three-dimensional variably-saturated subsurface flow solver (SERGHEI-RE)", the authors develop the Richards Equation-based variably-saturated subsurface flow module of SERGHEI. The characteristics of the framework are accuracy, robustness, scalability, and portability, which preserves the parallel performance and scalability across different HPC hardware. There are still certain changes and clarifications that the authors should address prior to publication. For these reasons, I believe that the manuscript can be accepted for publication by the GMD after minor revision. Below, I have some general comments for the authors.

**General comments:**

1 - Line #48 – 49, it would be better provide further explanation on how parallel performance depends on HPC hardware. For example, are the differences in the parallel computing scripts caused by the different HPC compilers, scheduler, or are the differences due to the storage methods (e.g. HPSS, GPFS file system) used by the HPC? This would help readers quickly understand the practical problems that this study aims to solve. If only the compilation issues related to CPU parallelization and GPU parallelization are addressed (line #59-60), please appropriately discuss the limitations of this study.

Response: lines 48-49 is about the concept of "performance portability". Broadly, performance portability means being able to run software on different hardware (portability) but ensuring that a consistently good performance can be achieved across the different hardware. There are major challenges to achieve this, in terms of algorithm design (i.e., algorithms which are suited for parallel hardware), programming language (i.e., standard high level languages can be easily compiled on CPUs, but hardware-specific languages such as CUDA are required for GPUs), software design (data structures, memory operations) and also optimising code, builds and configuration workflows. Our paper is concerned with all these, although much less on the latter one. Our paper is really concerned about performant-portable code of the numerical solver itself, therefore the main issues are code design and portable programming language.

The reviewer asks about the differences in parallel computing scripts. It is not fully clear what this means. If "script" means the source code of the main solver, the issues of programming language and code design are key. Our approach to this is to leverage on Kokkos, so that we write code that can be compiled both on CPUs and GPUs, depending on the selected Kokkos backend. If "script" means the

workflows to build the CPU/GPU binaries, or scripts to configure and run jobs, then we must say this paper is not really concerned with this topic. Pragmatically, we do solve some of these issues to be able to run our jobs, and this is reflected in the software release where we include the relevant workflows to build Kokkos and KokkosKernels together with our CMake workflow to build SERGHEI for both CPUs and GPUs. We also provide templates for job submission scripts.

We are not concerned at all in this work with different file systems (e.g., GPFS, HPSS), although it must be acknowledged that the target file system may play a role in the IO performance. Similarly, although we provide in our software templates to deal with SLURM job submissions, we do not this exhaustively, but rather as a hint on how to do this, and we do not provide templates for other scheduling software. We now communicate this explicitly in the text.

"There are additional challenges when porting software across HPC systems, related to supporting robust and general workflows to deal with different software stacks, filesystems, schedulers, and so on. We do not address these issues in this paper."

"In this manuscript, we focus on demonstrating its performance as a Richards solver only. Moreover, we focus on the performance-portability of the SERGHEI-RE solver, and purposely do not delve into the workflows required to build and run the software although minimal support of this is included in the software release."

2 - Line #124-129, it would be necessary to provide specific examples of the use of PC and MP schemes in SERGHEI-RE. This would make it more intuitive and help better understand the advanced nature of this research plan.

Response: We have added more descriptions of how the time stepping is determined for the PC and MP schemes. We hope this helps the reader to better understand our model strategy.

"For the PC scheme implemented in SERGHEI-RE, $\Delta t$ is adjusted based on the maximum change of water content in the previous step. If the maximum change of water content within a time step ($\Delta\theta_{max}$) is greater than an upper limit, $\Theta_{max}$, $\Delta t$ is reduced. If $\Delta\theta_{max} < \Theta_{min}$, $\Delta t$ is increased. $\Theta_{max}$ and $\Theta_{min}$ are user-defined thresholds. For the MP scheme, $\Delta t$ is adjusted based on the number of linearization iterations, similar to the solution in Hydrus (Simunek et al., 2013). If the number of linearization iteration required for convergence ($N_{iter}$) is greater than an upper limit, $N_{iterMax}$, $\Delta t$ is reduced. If $N_{iter} < N_{iterMin}$, $\Delta t$ is increased. $N_{iterMax}$ and $N_{iterMin}$ are user-defined thresholds. "

3 - Equation 8 could be illustrated with a diagram like Fig.2.

Response: We have prepared a figure to illustrate Eq. 8:

[Figure]

4 - The comparison figures of PC and MP show that, except for the 4 days simulation in Fig. 4, they almost completely overlap. The introduction section regarding these two different types of introductions cannot fully explain the outcome. Therefore, it is best to provide further explanation.

Response: Both PC and MP schemes have been widely used in the literature. In most cases, they produce very similar results as shown in Fig. 4. However, under certain soil water conditions (e.g., fast infiltration into dry soil), the MP scheme might be unable to converge. Moreover, the computational speeds are significantly different for the two schemes, which cannot be reflected from Fig. 4. We have added more explanation in the Introduction section to address the performance of the two schemes. We have also referred the readers to Section 2.2 for a more detailed explanation of the two schemes.

"In a comparison study between popular solution schemes, Li et al., 2024 showed that for some simulation scenarios, most numerical schemes achieved satisfactory results. For other scenarios, however, only specific schemes can produce reasonable simulation results. Generally, iterative schemes (such as the modified Picard scheme) adopt a greater time step size (dt), but they could fail to converge under certain soil types or soil moisture conditions. Non-iterative schemes, on the contrary, are typically easy to converge, but a smaller dt is required that increases the computational cost. A more detailed explanation of these two types of numerical schemes will be provided in Section 2.2."

5 - Figure 11, almost all evaporation points are zero after 50 days, this seems to be caused by a model code error. If not, please explain briefly.

Response: We have checked the data again. The evaporation data was obtained from the simulation results reported in the paper of He et al., 2018, in which they mentioned that the error of the simulated evaporation rate was greater for year 2015. However, for this test problem, soil evaporation plays a minor role compared to transpiration and irrigation. Thus, it does not affect model verification. We have explained the abnormal evaporation rate in the manuscript:

"These data are all provided in He et al., 2018. The rainfall and irrigation data were sampled in the field, whereas the evaporation and transpiration data were simulated. In 2015, the simulation error of soil evaporation increased, shown as the nearly-zero evaporation flux after day 50 (Fig. 11c). However, for this test problem, soil evaporation is relatively insignificant compared to other fluxes, and it has minor influence on soil moisture dynamics."

**Specific comments:**

1 - Figure 1, in this diagram, what is the difference between the dashed lines, solid lines, and those without borders?

Response: We have added explanations of different line styles in the figure caption:

"An overview of the SERGHEI model components modified from Caviedes-Voullieme et al. (2023). The solid lines indicate operational modules. The dashed lines indicate modules in the experimental stage. Modules without lines are still in the conceptual stage."

2 - Line #100, what do α and represent?

Response: We have added explanations of the van Genuchten soil parameters:

"It has been generally acknowledged that α is related to the air entry value and n reflects the particle size distribution (Vogel, 2001; Zhang et al., 2021), but different opinions also exist (van Lier and Pinheiro, 2018)."

Thank you again for your constructive comments.

Sincerely,
Zhi Li (Corresponding Author)